# Thoracic Ultrasound in Lung Transplantation—Insights in the Field

**DOI:** 10.3390/life13030695

**Published:** 2023-03-04

**Authors:** Hans Henrik Lawaetz Schultz, Jesper Rømhild Davidsen

**Affiliations:** 1Department of Cardiology, Section for Lung Transplantation, Copenhagen University Hospital, Rigshospitalet, 2100 Copenhagen, Denmark; 2South Danish Center for Interstitial Lung Diseases (SCILS), Department of Respiratory Medicine, Odense University Hospital, 5000 Odense, Denmark; 3Pulmo-Rheuma Frontline Center (PURE), Department of Respiratory Medicine, Odense University Hospital, 5000 Odense, Denmark; 4Odense Respiratory Research Unit (ODIN), Department of Clinical Research, University of Southern Denmark, 5230 Odense, Denmark

**Keywords:** thoracic ultrasound, lung transplantation, acute lung allograft dysfunction, chronic lung allograft dysfunction, bronchiolitis obliterans syndrome, restrictive allograft syndrome, ex vivo lung perfusion

## Abstract

The use of thoracic ultrasound (TUS) is a novel and dynamic diagnostic and monitoring modality that has shown remarkable advances within the last decade, with several published papers investigating its role within the field of lung transplantation. The aim of this current opinion review is to review the existing literature on the role of TUS in all stages of LTx, from in-donor lung evaluation to graft assessment on ex vivo lung perfusion and in the short- and long-term follow-up after LTx.

## 1. Introduction

Thoracic ultrasound (TUS) has gradually been accepted as a chest imaging modality with high diagnostic and monitoring accuracy for a variety of respiratory diseases and conditions and thus is being increasingly used by a broad range of clinical specialties within respiratory medicine, intensive care, and emergency medicine [1]. Not least, TUS has gained further popularity, secondary to its growing use during the COVID-19 pandemic as part of the risk stratification to predict respiratory failure and admission to intensive care units, including prognostication in those patients suffering from acute respiratory distress syndrome [2,3].

In a clinical setting, satisfactory TUS performance necessitates the theoretical and practical skills obtained from several available certified TUS education possibilities, where the involvement of or add-on simulation-based training is recommended to improve clinical performance [4,5]. Such training also contributes to understanding the commonly used terminology and knowledge of normal and pathological findings of the lung parenchyma, pleura, and thoracic wall, which is an essential prerequisite for the sufficient interpretation of TUS observations [6,7]. Several TUS scanning protocols with different scanning areas and zones are described, but the most frequently used in prospective clinical studies is the 14-zone approach, which accommodates the scanning of the thorax’s anterior, lateral, and posterior surfaces [8,9,10,11,12] (Figure 1).

Though TUS might be regarded as a novel modality for clinical guidance in the management of lung transplantation (LTx), its wide range of applications within the different fields of LTx has received increased recognition [13,14]. However, due to LTx being a rare condition, the evidence for TUS use in all stages of LTx is restricted to a few studies that are very often based on observations from a limited number of lung transplant recipients. Nonetheless, these observations qualify TUS as a pivotal and valid bedside tool to detect LTx-related conditions where intrathoracic pathological findings may appear [1]. The TUS observations may relate to the time from the LTx operation being conducted and why the course of LTx can be differentiated into the following stages: (1) a pre-LTx stage (e.g., where the quality of the donor lungs can be assessed prior to surgery); (2) a short-term post-operative LTx-stage; (3) a late or long-term post-operative LTx-stage. Common for the postoperative LTx stages is an ongoing surveillance need for the continuous assessment of both acute lung allograft dysfunction (ALAD) [15] and chronic lung allograft dysfunction (CLAD) [16]. Generally, TUS is believed to possess still unused and unexploited potential in the LTx setting, which indirectly calls for TUS’s benefits to be clarified.

## 2. Aim

The aim of this current opinion review is to review the existing literature on the role of TUS in all stages of LTx, from in-donor lung evaluation to graft assessment on ex vivo lung perfusion and in the short- and long-term follow-up after LTx.

## 3. Knowledge of TUS and LTx

### 3.1. The Use of TUS in Pre-LTx Stages

The evaluation of lungs for LTx, using TUS, both “in-donor” and on an ex vivo lung perfusion system (EVLP), has been attempted in different settings. In a small case series that included six potential donors, Lebovitz et al. performed “in-donor” lung evaluation using TUS on neurologically deceased donors and compared the TUS findings with the findings on conventional chest X-rays in anterior and posterior planes. Of the six neurologically deceased potential donors, five of them were used for LTx, whereas one was unsuitable. They found that 6/6 evaluated potential donors had bilateral consolidations, pleural effusion was observed in 4/6 potential donors, and edema was seen in 5/6 potential donors. In contrast, a conventional chest X-ray found consolidations/atelectasis in 3/6 of potential donors, pleural effusion in 1/6 potential donors and no edema, and on the basis of these findings, it was concluded that TUS might be a more accurate and efficient tool to evaluate lungs for potential donation [17]. To our knowledge, no other studies have been published on this subject, but the potential utility of TUS to rule out pleural effusion and edema, more in a pre-LTx setting, is greater than conventional X-ray. TUS can also be used to track dynamic changes over time in a donor management situation.

EVLP is a rapidly growing procurement technique used worldwide to evaluate the suitability of donor lungs. The potential donor lungs are removed from the donor and placed on the EVLP system, allowing transplant surgeons and physicians to evaluate the suitability of the donor organ for LTx in marginal donors. In a recently published study, Ayyat et al. performed an ultrasound on 45 lungs from 23 donors placed on an EVLP system to evaluate any applied therapeutic approach during EVLP [18]. A CLUE (direCt Lung Ultrasound Evaluation) score was developed as the number of Grade 1 images × 1 number of Grade 2 images × 2+ numbers of Grade 3 images × 3+ numbers of Grade 4 images × 4+ numbers of consolidation images × 5 in relation to the total number of images taken. Grade 0 was defined as no B-lines on TUS; grade 1 was defined as 1–25% B-lines; grade 2 was defined as 26–50% B-lines; grade 3 was defined as 51–75% B-lines; and grade 4 was defined as a 76–100% B-line coverage or white-out (i.e., coalescence of B-lines generating a typical white lung image). The final total CLUE score had the highest area under the curve (AUC) of 0.98 (sensitivity 100% and specificity 86%) compared with other EVLP evaluation parameters, such as the ratio of arterial oxygen partial pressure (i.e., PaO_2_ in mmHg) to fractional inspired oxygen (i.e., P/F ratio) of 0.58, a change in PaO_2_ of 0.61, pulmonary artery pressure of 0.71, pulmonary vascular resistance of 0.64, peak airway pressure of 0.83, and dynamic compliance of 0.85. The TUS-related CLUE scores were calculated and used as adjunctive information for the final consensus decision on suitability for LTx, and therefore, the CLUE score actively impacted the suitability decision, which is a limitation of this study. Post-transplantation follow-up was not reported in this paper.

### 3.2. TUS Use in Short-Term Post-LTx Stages

Acute lung allograft dysfunction (ALAD) refers to a condition of acute graft dysfunction, presenting in the early course after LTx with an often-used time window of within 90 days in which the graft only retains marginal function [19]. ALAD has a different etiology; however, immediate allograft dysfunction may develop almost in prolongation of the operative procedure, and in up to 25–50% of cases, the underlying cause relates to primary graft dysfunction (PGD), defined as an acute non-immune-mediated injury to the transplanted lung occurring within the first 72 h postoperatively and which is caused by a combination of ischemia, reperfusion, cold organ preservation, and pretransplant pulmonary hypertension [20,21]. Other early (>72 h) risk factors or conditions in the short-term post-LTx stage leading to ALAD comprise acute cellular rejection (ACR), mechanical abnormalities (anastomotic strictures, bronchomalacia, diaphragm paralysis, and pleural effusion, including hemothorax), pneumonia (viral, bacterial, and fungal infections), thromboembolic disease, anemia, and gastro-esophageal reflux [15,22,23,24,25]. In the immediate post-transplantation setting, the use of computed tomography (CT) and other radiological methods is often challenged by a sedated ICU patient with bandages, chest tubes, and catheters, and although TUS might be challenging, it has the advantage of a bedside approach.

The use of TUS to discover the presence of many of the mentioned underlying conditions for developing ALAD has been explored in one national follow-up study from Denmark, where Davidsen et al. prospectively investigated 14 lung transplant recipients (13/1 double LTX (DLTx)/single LTx (SLTx)) at four time-points corresponding to post-LTx day 3 (TUS #1), 14 (TUS #2), 42 (TUS #3), and 84 (TUS #4) [11]. In this study, the most frequent pathological finding identified by TUS was pleural effusion, observed in 85.7%, 92.9%, 85.7%, and 78.6% of the lung transplant recipients equivalent to TUS#1-4, which may imitate a high prevalence throughout the entire 84-days observation period (Figure 2). However, in the study, the presence of pleural effusion was dichotomized, and as TUS can detect pleural effusion below 20 milliliters, this likely explained the findings that are consistent with an overall decreasing presentation over time, going from bilateral to unilateral and large/moderate to small pleural effusion. At TUS#2, the highest prevalence of compression atelectasis was found to be reliable, with the highest prevalence of pleural effusion. Pneumonia was most predominant at TUS#2 (28.6%), with decreasing prevalence at TUS#3-4 (14.3%), and during the observation period, neither lung transplant recipients were diagnosed with pulmonary embolism nor interstitial syndrome. Still, the most profound finding was that TUS as an add-on modality to ordinary LTx surveillance had a clinical impact in 10/14 lung transplant recipients (71.4%) during the observation period due to the detection of diagnoses with diverting interventions, such as re-operation due to sternotomy-related wound infection, pleural drainage, and initiation of/or changed antibiotic strategy. The prevalence of TUS-induced interventions was highest at post-transplant day 14 (TUS#2) and comprised half of the lung transplant recipients (*n* = 7 (50%)). The TUS observed findings from this study were assumed to be representative and compatible with the available knowledge on time-dependent pulmonary complications in the short-term post-LTx stages [26].

### 3.3. TUS Use in Long-Term Post-LTx Stages

The majority of lung transplant recipients will have obtained close to maximal post-LTx lung function around one year after LTx [27]. Nonetheless, underlying conditions, as described for ALAD, may still complicate the LTx course, giving rise to CLAD, which occurs with increasing prevalence one year after LTx, supporting the demand for long-term follow-up [16,22,28]. CLAD is defined as a persistent decline (>three months) in forced expiratory volume in one second (FEV1) of >20% compared to the best post-transplant baseline value. This is further phenotyped into either bronchiolitis obliterans syndrome (BOS) with an obstructive ventilation pattern, restricted allograft syndrome (RAS) with a restrictive ventilation pattern (i.e., a decline in total lung capacity (TLC) of < 90% of the best post-transplant baseline value) in combination with the presence of pleura-parenchymal opacities on a high-resolution computed tomography (HRCT), which is often manifested as apical pleura-parenchymal fibroelastosis (PPFE), or a mixed-type CLAD [16,28,29].

One prospective French study by Droneau et al. aimed to validate whether TUS could be used in a long-term post-LTx setting, with the performance of TUS in lung transplant recipients who had undergone LTx on an average of 23 months prior to inclusion (median: 9 months) [30]. The study concluded that an ordinary TUS approach, including standard variables (e.g., the presence of B-lines, lung sliding, lung pulse, and seashore sign), was feasible in lung transplant recipients. However, as the cohort comprised 22 presumable lung healthy transplant recipients, most TUS findings were expectedly normal, which is why it was emphasized that using TUS needed further exploration in long-term post-LTx settings.

In 2017, Davidsen et al. published a case report proposing that TUS could be used as a novel tool to phenotype CLAD [31]. The same study group further investigated this hypothesis in a later observational study with the prospective inclusion of 25 lung transplant recipients with new-onset CLAD (n(BOS):n(RAS) = 19:6), who were examined with TUS and HRCT, performed within an average time window of ten days prior to or after TUS [12]. HRCT was used as the gold standard for the PPFE findings that would correspond to the TUS findings demonstrating pleural thickening. It was found that the RAS patients were presenting differently from the BOS patients, with a significant difference in pleural thickening, as measured by a TUS of 5.6 mm and 2.9 mm (normal pleura thickness is 1 mm), respectively, and was consistent with a significantly higher prevalence of PPFE findings on HRCT in the RAS compared to BOS patients. Importantly, the study proved a high diagnostic accuracy of TUS to identify apical pleural thickening in either the anterior or posterior apical zones in RAS patients as a surrogate marker of HRCT-verified PPFE in lung transplant recipients (i.e., with a sensitivity of 100% (95% CI; 54–100%), specificity of 100% (95% CI; 82–100%), positive predictive value (PPV) of 100% (95% CI; 54–100%), and a negative predictive value (NPV) of 100% (95% CI; 82–100%)) (Figure 3). Hence, confirmation of this novel so-called RAS-sign in lung transplant recipients with developed CLAD increased the probability of RAS over BOS. As such, it was concluded that TUS could discriminate RAS from BOS and thus be used as an up-front tool for CLAD phenotyping, which clinically impacts the reduction of time for diagnosing RAS, which is crucial since its prognosis is inferior to BOS [32,33,34].

### 3.4. Other Associations to LTx

Various studies have attempted to describe the association between ACR and computed tomography (CT) abnormalities. For example, Gotway et al. found limited accuracy in the use of CT to diagnose ACR [35], while Park et al. found an association between ACR and bilateral ground-glass opacities and septal thickening with lower predominance, which has not been proven afterwards [36]. However, no studies have evaluated TUS as a possible method to diagnose ACR, which might be due to the divergent results and the limited accuracy of CT in diagnosing ACR.

Post-procedure pneumothorax identification in lung transplant recipients has been investigated in one study by Bensted et al. [37]. They performed TUS on 165 patients and found eight pneumothoraces after conducting transbronchial biopsies. Using the chest X-ray as the gold standard in their study, TUS was shown to have a sensitivity of 75%, specificity of 93%, PPV of 35%, and NPV of 99% in their study. These results parallel the findings from a previous meta-analysis on TUS being used to detect pneumothorax [38].

## 4. TUS Future Research—Advantages and Disadvantages of TUS

Despite the fact that only a few TUS studies are carried out within the field of LTx, TUS has manifested itself as a valuable and indispensable bedside tool to detect and monitor pathological conditions in relation to LTx, in addition to being used as a tool with impact to influence further clinical decision making prior to or after LTx. In this context, TUS can be used to exclude LTx-related conditions/complications, thereby avoiding the waste of otherwise unnecessary investigations required to clarify a clinical suspicion. Still, more studies are required on larger LTx cohorts and with longer observation periods (>24 months post-LTx) to obtain a more profound understanding and acquire more qualified data on when, and on which focus areas TUS should be used in the LTx setting, considering the dynamic pathological changes and exposures (e.g., LTx immunosuppression) that will inevitably appear over time. In this perspective, gold standard examinations (preferable chest CT/HRCT) should optimally be performed alongside TUS to achieve the most reliable comparison and to prevent under- or over-diagnostics of pathological findings/LTx-complications that may disappear or appear when accepting a long time window between TUS and gold standard testing. Based on such knowledge, future studies should focus on TUS as an integrated part of short-term as well as long-term LTx surveillance follow-up. This should aim to investigate whether TUS implications may optimize the earlier detection of LTx-complications and avoid laborious follow-up set-ups, including CT, which would be in favor of lung transplant recipients but also regarding a reduction in health care costs (e.g., an RCT comparing follow-up, including TUS vs. a standard follow-up). TUS use for the selection of potential LTx candidates has not been investigated, but it presumably could contribute to excluding candidates with obvious intrathoracic pathology, such as severe PPFE with lung shrinkage with pleural adhesions and thickening, which would challenge LTx surgery.

Importantly, TUS should be regarded as an adjunction to the existing examinations regarding selection for and follow-up of LTx, including phenotyping CLAD, and cannot stand alone. However, the immediate advantages and disadvantages of TUS are illustrated in Table 1.

## 5. Conclusions

TUS has a huge potential within the field of diagnostics and monitoring of more respiratory diseases, and recently its use has also been shown to have a clinical impact on rare respiratory conditions, such as LTx. Several studies have investigated the use of TUS in different stages of LTx. When evaluating “in-donor” lungs, TUS is a helpful tool with a higher accuracy than conventional chest X-rays. When using the ex vivo evaluation of lungs for transplantation, TUS is a possible add-on examination with a higher AUC than any of the other evaluation methods used in the EVLP systems; however, it was more labor-intensive. In the immediate postoperative course, TUS offers many advantages and convenience as a point-of-care examination and effectively identifies early complications, such as pleural effusion, consolidation, and pneumothorax. In a long-term follow-up and in the phenotyping of CLAD, TUS has demonstrated its potential as a suitable screening tool with high accuracy when in the hands of trained experts. In conclusion, TUS is believed to possess still unused and unexploited potential in the LTx setting, which indirectly calls for TUS’s benefits to be clarified in future large multicenter and randomized studies.

## Figures and Tables

**Figure 1 life-13-00695-f001:**
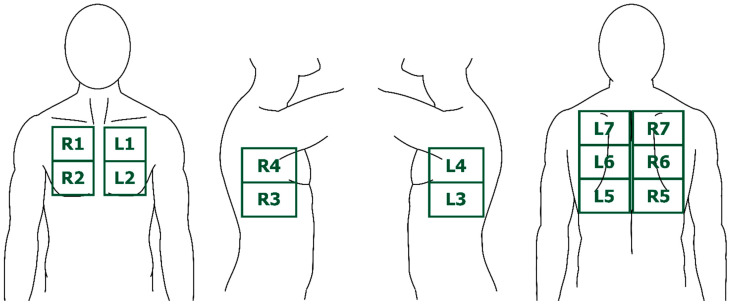
The seven LUS zones on each hemithorax, corresponding to the anterior, lateral, and posterior thorax walls. Abbreviations: L = left. R = right (this figure was originally published in Davidsen et al. [12]).

**Figure 2 life-13-00695-f002:**
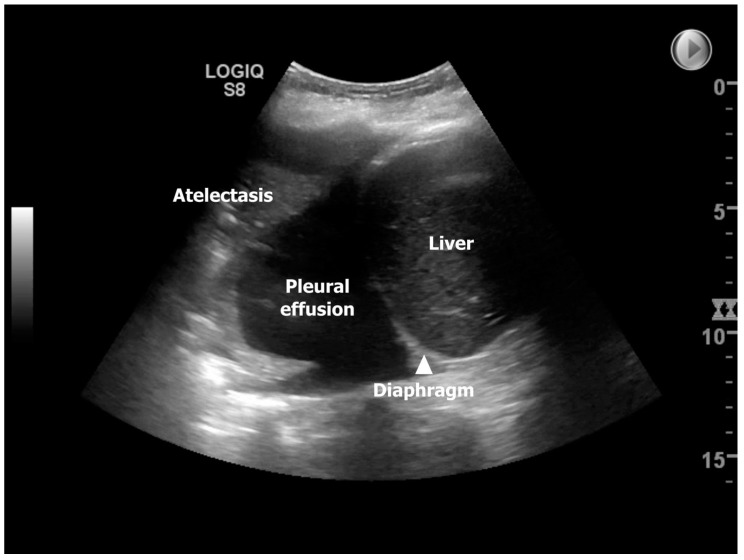
Showing an ultrasound image in zone 3R on day 14. Ultrasonographic findings from a patient corresponding to post-transplant day 14 illustrating a moderate right-sided pleura effusion with concurrent compression atelectasis [11].

**Figure 3 life-13-00695-f003:**
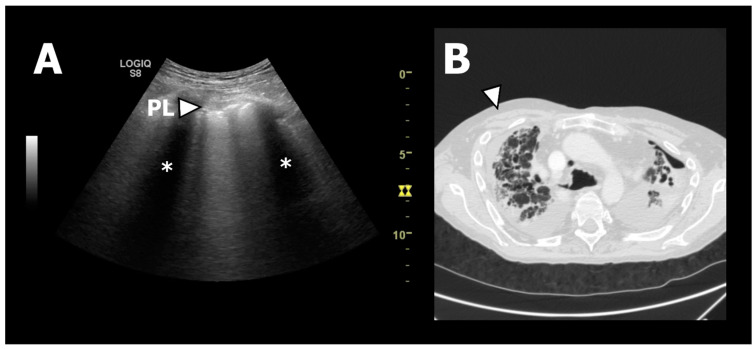
TUS and HRCT findings from a patient with RAS. (**A**): The white arrow indicates TUS from zone R1 showing a thickened and fragmented pleural line. * indicates the subpleural “shadowing” corresponding to the ribs. (**B**): Axial HRCT image of the upper lobes showing fibrotic pleural and septal thickening consistent with PPFE. The white arrow corresponds to TUS zone R1 presented in (**A**) [12].

**Table 1 life-13-00695-t001:** Advantages and disadvantages of TUS and HRCT in the LTx setting.

	Advantages	Disadvantages
TUS	Minimal time-consuming	Interobserver variability
	Cheap	
	Easy feasible	
	Frequent monitoring possible	
	Radiation-free	
	High accuracy for parenchymal and pleural pathology	
	Pathological findings contribute to clinical decision making	
	Can exclude diseases/complications to LTx	
Regarding CLAD	May trigger further CLAD examination by identification of the RAS-sign	Cannot identify BOS
	Ruling out other pathological findings not related to CLAD	Only suggestive of PPFE-pattern compatible with RAS and/or mixed-type CLAD
HRCT	Can quantify pathological pleural and parenchymal affection	Radiation risk
	Can confirm diagnosis/condition/complication	Time-consuming
	Acts as a gold standard in many aspects of respiratory diseases/conditions	Resource-heavy (referral and logistics)
	Detailed illustration of the entire thoracic anatomy	Expensive
Regarding CLAD	Ruling out other pathological findings not related to CLAD Confirmative of radiological RAS and BOS patterns	
MRI	Radiation-free thoracic imaging modality visualizing thoracic anatomy in detail.	No available evidence on this field within LTx and overall spare experience of the value of MRI for diagnosing and following up pulmonary diseases, not least how MRI and its different phase recordings should be interpreted

Parts of this table were originally published as Supplemental Material in [12].

## Data Availability

No new data were created or analyzed in this study. Data sharing is not applicable to this article.

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
