# Peer review of "Thoracic Ultrasound in Lung Transplantation—Insights in the Field"

_life, 2023, doi:10.3390/life13030695_

Round 1

Reviewer 1 Report

The manuscript is little difficult to follow in several paragraphs and could benefit with some minor editing.

Donor section- TUS picked up more abnormalities in all donor lungs (6/6) rather than CXRs (3/6 abnormal) but is it too sensitive since 5/6 donor lungs were used for transplant.  It is not clear about the outcomes or if there was a grading system?  The authors could comment on its potential utility.

EVLP – TUS had the highest sensitivity and specificity compared to other parameters, but it is not clear what this relates to or the defining of lungs that can be used for transplant.  Also, not clear as to what was the ultimate outcome of the donor lungs. 

In the use of TUS in the short term section:  

It appears that TUS could be useful early in the course, possibly for PGD since as noted in the introduction, TUS is useful for ARDS with COVID, although not sure how practical or useful with chest tubes and bandages in place. 

Day 3, 14, 42, and 84 study:

Day 14—per report appears to be the time in which there were the most pathologic findings.  It would be of interest, if these findings impacted care or not.  It is interesting that, the presence of pleural effusion was seen in most patients even out to day 84.  Again, would be interesting the impact of this on care. 

It does not sound like there is information regarding the use of TUS for acute rejection (ACR or AMR), however, the authors may want to comment on whether the use of TUS could be helpful in distinguishing other causes of reduced lung function rather than ACR such as pneumonia vs ACR, 

Figure 2 Wrong title and it is the same as figure 1.

TUS appears to be good to differentiate BOS vs RAS (oCLAD – rCLAD) with good diagnostic differences.  However the figure illustrating this is quite severe pleural thickening.  One question would be how early this can be identified with TUS.   

Conclusions --- AOC should be AUC

Reviewer 2 Report

In the manuscript “Thoracic ultrasound in lung transplantation – insights in the field”, Schultz and Davidsen provided an opinion review regarding the use of thoracic ultrasound for assessments in lung transplant recipients.

Specific comments:

1.     Overall, the manuscript is well-written and summarizes evidence of the utility of thoracic ultrasound for assessing lung transplant recipients. Because there are only a few studies on this subject thus far, the authors aimed to cover most publications on it.

2.     There is some type throughout the manuscript, which can be easily corrected by a new round of verification.

3.     The aim should be rephrased. Perhaps the sentence can be divided in two to facilitate the reading.

4.     Advantages and disadvantages of MRI should also be included in Table 1.
